# Neuronal sources of *hedgehog* modulate neurogenesis in the adult planarian brain

Ko W Currie[1,2], Alyssa M Molinaro[1,2], Bret J Pearson[1,2,3]*

[1]Program in Developmental and Stem Cell Biology, Hospital for Sick Children, Toronto, Canada; [2]Department of Molecular Genetics, University of Toronto, Toronto, Canada; [3]Ontario Institute for Cancer Research, Toronto, Canada

**Abstract** The asexual freshwater planarian is a constitutive adult, whose central nervous system (CNS) is in a state of constant homeostatic neurogenesis. However, very little is known about the extrinsic signals that act on planarian stem cells to modulate rates of neurogenesis. We have identified two planarian homeobox transcription factors, *Smed-nkx2.1* and *Smed-arx*, which are required for the maintenance of cholinergic, GABAergic, and octopaminergic neurons in the planarian CNS. These very same neurons also produce the planarian *hedgehog* ligand (*Smed-hh*), which appears to communicate with brain-adjacent stem cells to promote normal levels of neurogenesis. Planarian stem cells nearby the brain express core *hh* signal transduction genes, and consistent *hh* signaling levels are required to maintain normal production of neural progenitor cells and new mature cholinergic neurons, revealing an important mitogenic role for the planarian *hh* signaling molecule in the adult CNS.

## Introduction

Once thought to be a non-existent phenomenon, homeostatic adult neurogenesis is a common trait shared by many disparate organisms including rodents, birds, flies, and humans (*Altman, 1962, 1969*; *Doetsch et al., 1999*; *Goldman and Nottebohm, 1983*; *Eriksson et al., 1998*; *Fernández-Hernández et al., 2013*). However, levels of neuronal turnover are tightly limited in these animals (*Obernier et al., 2014*). In fact, the highest known site of adult homeostatic neurogenesis in the human central nervous system (CNS) is the hippocampus, where annual neuronal turnover rates are estimated to be only 1.75% (*Spalding et al., 2013*; *Sanai et al., 2011*; *Bergmann et al., 2012*). Adult neurogenesis in most animals depends on the action of ectodermally derived neural stem cells, which have radial-glial character and are integrated into a stable niche microenvironment. Extrinsic signals such as wingless (Wnt), sonic hedgehog (Shh), and bone morphogenetic proteins (BMPs) act to finely control neural stem cell proliferation and differentiation (*Silva-Vargas et al., 2013*; *Lehtinen et al., 2011*).

The asexual strain of the freshwater planarian, *Schmidtea mediterranea* (*S. mediterranea*), is a constitutive adult flatworm (Lophotrochozoan) that challenges the dogma of low rates of adult neurogenesis. Not only does *S. mediterranea* constantly turnover its brain, but also it is capable of complete brain regeneration within only two weeks following decapitation (*Cebria, 2007*; *Reddien and Sánchez Alvarado, 2004*; *Newmark and Sánchez Alvarado, 2002*). In addition, the uninjured planarian CNS is known to be a highly dynamic organ, which can adjust its size through the addition or subtraction of mature neurons to maintain consistent proportions with the rest of the body as it grows and shrinks, respectively (*Baguñá and Romero, 1981*; *Hill and Petersen, 2015*). Amazingly, these regenerative feats and high levels of homeostatic neurogenesis are accomplished in the absence of a recognizable neuroepithelium, and without any definitive neural stem cells (*van Wolfswinkel et al., 2014*; *Zhu et al., 2015*). Recently, brain-derived Wnt signals have been

*For correspondence: bret.
pearson@utoronto.ca

**Competing interests:** The authors declare that no competing interests exist.

**eLife digest** Most animals can continue to generate and add new neurons in their nervous system into adulthood, though the process is often tightly regulated. In adult humans, only a small number of neurons are made or lost, such that the fewer than 2% of the neurons in the nervous will change over, or "turnover", the course of a year.

The turnover of neurons in some other animals is much higher than it is in humans. A freshwater flatworm, called *Schmidtea mediterranea*, is one example of such an animal that can even regenerate an entirely new brain if its head is decapitated. These flatworms have a large population of adult stem cells, which makes these high rates of neuron production and regeneration possible. However, it is largely unknown if this population contains stem cells that can only become new neurons, in other words "dedicated neuronal stem cells". Moreover, it is also not clear what kinds of signals communicate with these stem cells to promote the production of new neurons.

In animals from flies to humans, a signaling molecule encoded by a gene called *hedgehog* forms part of a signaling pathway that can promote neuron production during development. Therefore, Currie et al. asked if the *hedgehog* signaling molecule might communicate with the stem cells in adult flatworms to control how many new neurons they produce.

The experiments revealed that the *hedgehog* signaling molecule is almost exclusively produced by the flatworm's brain and the pair of nerve cords that run the length of the flatworm. Currie et al. then found a smaller group of cells close to the flatworm's brain that looked like dedicated neural stem cells. These cells can receive the *hedgehog* signals, and further experiments showed that flatworm's brain requires *hedgehog* signaling to be able to produce new neurons at its normal level.

The *hedgehog* signaling molecule is likely only one of many signaling molecules that regulate the production of new neurons in flatworms. It will be important to uncover these additional signals and understand how they work in concert. In the future, a better understanding of this process will help efforts to devise ways to induce humans to replace neurons that are lost following injury or neurodegenerative diseases.

shown to influence the neurogenic output of planarian stem cells (neoblasts) during regeneration (*Hill and Petersen, 2015*). However, little is known about the specific extracellular signals and transcription factors that modulate neoblast activity within this body region to balance cell proliferation and neuronal differentiation, which undoubtedly involves many overlapping regulatory systems.

Here, we have identified two planarian homeodomain transcription factors, *Smed-nkx2.1* and *Smed-arx* (henceforth referred to as *nkx2.1* and *arx*), which act to specify cholinergic, GABAergic, and octopaminergic neurons within the ventral-medial (VM) region of the planarian CNS. Interestingly, these very same VM neural cell types are a major source of the planarian hedgehog ligand (*Smed-hh*; henceforth referred to as *hh*) (*Rink et al., 2009*; *Yazawa et al., 2009*). We also describe a novel neoblast population, which are *nkx2.1*[+] or *arx*[+], express the core machinery for the reception of *hh* and Wnt signals, and are located adjacent to *hh*[+] VM neurons. Finally, we present evidence that consistent *hh* signaling within this neoblast microenvironment is required to promote normal homeostatic neurogenesis of the VM neuronal population. In total, we identify a *hh* signaling axis that positively modulates VM neurogenesis through distinct progenitor cells.

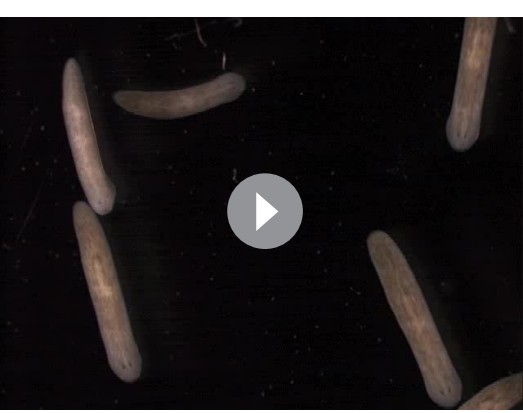

**Video 1.** *Control(RNAi)* worms exhibiting normal locomotion and ability to flip over from dorsal surface back onto ventral surface.

## Results

### *nkx2.1* and *arx* are expressed in ventral-medial neural cell types

*nkx2.1* and *arx* were originally cloned and isolated during an RNAi screen aimed at identifying planarian transcription factors with potential roles in neuronal specification. A unique behavioral defect was observed in all *nkx2.1(RNAi)* animals, characterized by tonic muscular contractions that bend the head dorsally, so that it is perpendicular with the rest of the body. Interestingly, some of these *nkx2.1(RNAi)* worms end up on their lateral body edge, causing these animals to move in tight circles, compared to *control(RNAi)* animals which move in straight lines on their ventral surface (*Videos 1* and *2*). In contrast, *arx(RNAi)* animals do not display overt behavioral defects, except when flipped onto their dorsal side where these worms have difficulty to corkscrew in order to reorient onto their ventral surface compared to *control(RNAi)* and wild-type animals (*Videos 1* and *3*).

In order to investigate the root cause of these behavioral defects, and whether *nkx2.1* and *arx* might function in neuronal specification and maintenance, whole-mount in situ hybridizations (WISH) and cross-sections were first performed on wild-type animals. *nkx2.1* exhibited broad expression throughout the body but was clearly present within the CNS (black arrow), whereas *arx* expression was spatially restricted to the medial CNS, including the anterior brain and the full length of the ventral nerve cords (*Figure 1A*). Transverse sections through the anterior-posterior (A-P) midpoint of the brain enabled imaging in greater spatial detail and highlighted the concentrated expression of *arx* within the ventral-medial region of each brain lobe (*Figure 1B*). In addition, expression of *nkx2.1* and *arx* was observed within a few isolated cells in between the brain lobes, a region largely populated by adult stem cells (*Figure 1B*, yellow arrows).

In order to determine which neuronal subtypes express *nkx2.1* and *arx*, double fluorescent in situ hybridization (dFISH) was performed, comparing either of these two transcription factors with established markers of either cholinergic, GABAergic, octopaminergic, dopaminergic, or serotonergic neurons (*Nishimura et al., 2007a*, *2007b*, *2008a*, *2008b*, *2010*). Both *nkx2.1* and *arx* were found to be expressed by neural cell types that occupy the VM region of the planarian brain (cholinergic, GABAergic, and octopaminergic neurons) but were largely excluded from dopaminergic and serotonergic neurons (*Figure 1C–E*; *Figure 1—figure supplement 1*).

Cholinergic neurons are by far the most populous cell type in the planarian CNS and are densely distributed throughout the brain (*Figure 1C*), making it difficult to assess the level of expression of *nkx2.1* and *arx* within this entire neuronal population. Therefore, a more focused region of interest was imaged to quantify the degree of expression of *nkx2.1* and *arx* within cholinergic neurons (*chat*[+]) (*Nishimura et al., 2010*). Transverse sections were first taken at the A-P midpoint of the brain, and individual brain lobes were imaged. Each lobe was then divided into a VM and dorsal-lateral (DL) half, with the line of division occurring between the widest point of each brain lobe (*Figure 1C*; yellow line). A considerable proportion of cholinergic neurons expressed *nkx2.1* in both the VM (36.89 ± 5.46%) and DL (23.33 ± 5.13%) brain regions (*Figure 1C*). In contrast, *arx* expression was more heavily biased toward the VM brain region, where 63.21 ± 4.35% of cholinergic neurons expressed this transcription factor (*Figure 1C*).

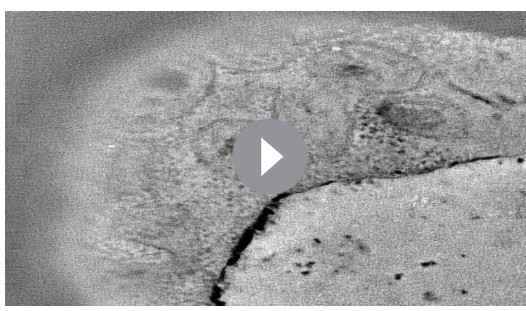

**Video 2.** *nkx2.1(RNAi)* worms exhibiting abnormal behaviors, including tonic muscular contractions.

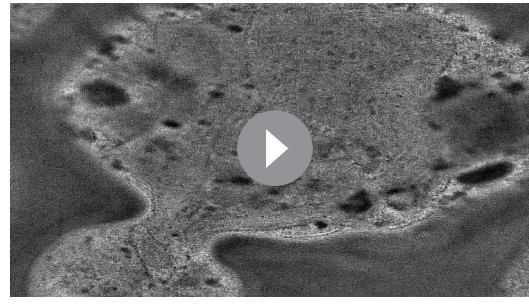

**Video 3.** *arx(RNAi)* worms exhibiting abnormal muscular contractions and inability to flip back onto dorsal surface.

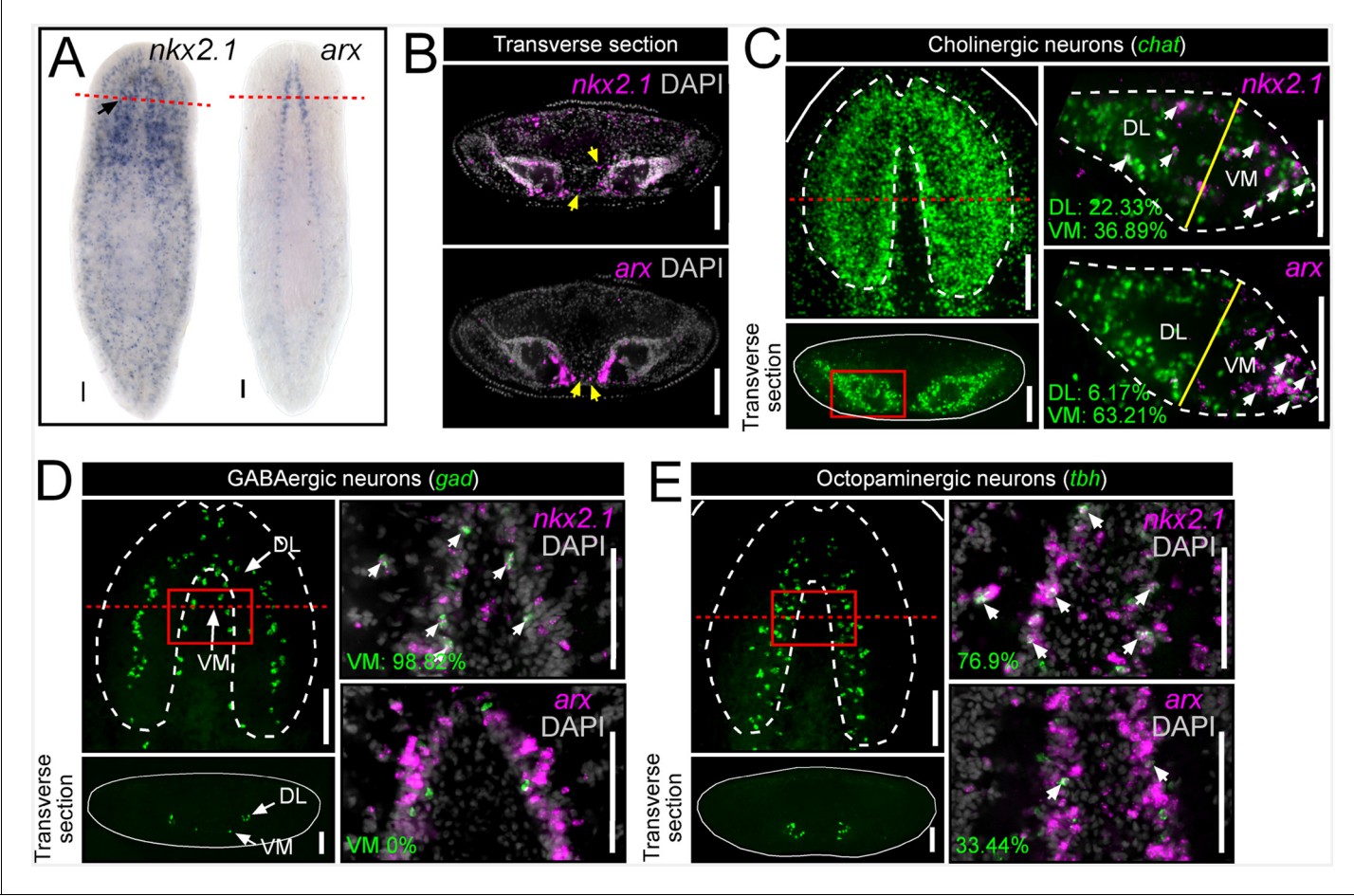

**Figure 1.** *nkx2.1* and *arx* are expressed by VM neural cell types. (**A**) WISH images of *nkx2.1* and *arx* expression. Black arrow highlights CNS expression. (**B**) Single confocal plane from transverse sections of FISH images depicting expression of *nkx2.1* and *arx* within the CNS. Yellow arrows highlight *nkx2.1* and *arx* expression within stem cell-rich area. (**C–E**) For each set of images, the upper left panels are projections of confocal images from FISH experiments showing all cholinergic, GABAergic, or octopaminergic neurons in wild-type animals. Solid red boxes represent region of interest for the right panels, which are single confocal planes from dFISH experiments, showing co-expression of *nkx2.1* or *arx* within a given neuron subtype. Solid yellow lines in (**C**) demarcate the division line between the VM and DL brain regions. White arrows in dFISH panels indicate double-positive cells. Solid white lines represent the border of the animal. Dashed white lines represent the approximate border of the CNS. Red dashed lines indicate approximate plane for transverse sections. Scale bar = 100 µm. CNS, Central nervous system; DL, Dorsal-lateral; FISH, Fluorescent in situ hybridization; VM, Ventral-medial.

The following figure supplement is available for figure 1:

**Figure supplement 1.** *nkx2.1* and *arx* are not expressed by DL GABAergic, dopaminergic, or serotonergic neurons.

Planarian GABAergic neurons (*gad*[+]) are present in two spatially distinct VM and DL subgroups (*Figure 1D*) (*Nishimura et al., 2008b*). Within the VM GABAergic neuron population, virtually all cells that were imaged expressed *nkx2.1* (98.82 ± 2.63%), whereas expression of *arx* was not detected (*Figure 1D*). In contrast, the DL subgroup of GABAergic neurons did not exhibit significant expression of either transcription factor (*Figure 1—figure supplement 1*).

Due to the relatively modest size of the octopaminergic neuronal subpopulation (*tbh*[+]), and the focal grouping of these cells within the medial region of the CNS, this neural subtype was assessed as a whole. Interestingly, expression of both transcription factors was observed within this neural cell group, with 76.9 ± 6.66% of octopaminergic neurons exhibiting *nkx2.1* expression, while a lower proportion were *arx*[+] (33.44 ± 6.54%) (*Figure 1E*). Taken together, these expression data suggested a

potential role for *nkx2.1* and *arx* in the specification and/or maintenance of VM cholinergic, GABAergic, and octopaminergic neurons.

## *nkx2.1* and *arx* are required for the maintenance of VM neural cell types

In order to test the functional roles in neuronal specification, *nkx2.1* and *arx* were knocked down by RNAi, and the cholinergic, GABAergic, and octopaminergic neuronal subtypes were assayed for any changes to cell number during homeostasis only (see Materials and methods). Compared to *control (RNAi)* animals, a loss of *arx* resulted in a visible reduction of cholinergic neurons in the VM region of the CNS (*Figure 2A*; white arrows), while *nkx2.1(RNAi)* worms appeared relatively unaffected (*Figure 2—figure supplement 1*). In order to quantify the loss of cholinergic neurons in *arx(RNAi)* animals, transverse sections of single brain lobes were imaged as described above, and 20 µm of

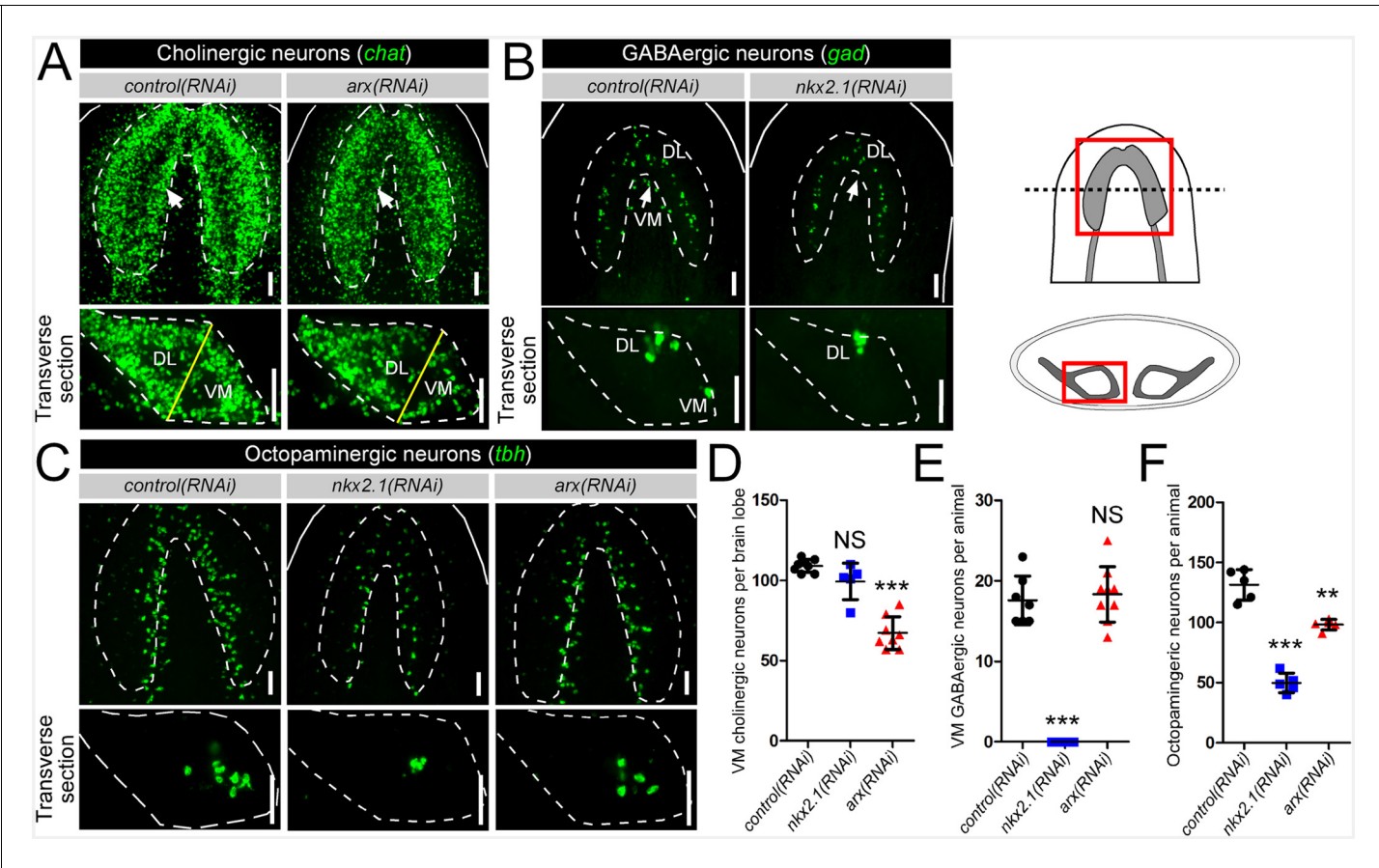

**Figure 2.** *nkx2.1* and *arx* are required for the maintenance of VM neural cell types. (**A–C**) Projected confocal image stacks showing cholinergic, GABAergic, or octopaminergic neuronal populations in RNAi-treated animals. VM neural subpopulations are highlighted by white arrows. Regions of interest are represented by cartoons at the right side of the figure. Solid white lines represent the outer border of the animal, and dashed white lines mark the outer border of the CNS. Solid yellow line demarcates the border between the VM and DL brain regions. Scale bar = 50 µm. (**D–F**) Quantification of VM cholinergic, VM GABAergic, and octopaminergic neurons in RNAi-treated animals. Significance levels in dot plots: **p<0.01, ***p<0.001, error bars are standard deviation. CNS, Central nervous system; DL, Dorsal-lateral; VM, Ventral-medial.

The following figure supplements are available for figure 2:

**Figure supplement 1.** *nkx2.1* and *arx* are not required for maintenance of DL neural cell types.

**Figure supplement 2.** *nkx2.1* and *arx* are not required for the maintenance of dopaminergic or serotonergic neurons Projected confocal images showing dopaminergic neurons or serotonergic neurons.

confocal depth was recorded. Subsequently, the number of *chat*[+] cholinergic neurons were counted within each of the VM and DL brain regions. Strikingly, *arx(RNAi)* worms exhibited a near 40% reduction in the number of VM cholinergic neurons, whereas DL neurons were largely unchanged (*Figure 2A,D*, *Figure 2—figure supplement 1*). In contrast, *nxk2.1(RNAi)* worms displayed no detectable changes in the average number of both VM and DL cholinergic neurons (*Figure 2—figure supplement 1*). This very specific loss of VM cholinergic neurons in *arx(RNAi)* worms, combined with the evidence that *arx* was expressed by >60% of these neurons, demonstrated that *arx* was required for the maintenance of VM cholinergic neurons.

*nkx2.1* represented an excellent candidate to act in the specification of VM GABAergic neurons based on its expression within nearly all of these cells. As there are many fewer GABAergic than cholinergic neurons in the planarian brain, imaging and quantification of this entire neuronal population was performed. As expected, *arx(RNAi)* animals exhibited no loss of GABAergic neurons in either the VM or the DL subgroups (*Figure 2B,E*, *Figure 2—figure supplement 1*). In contrast, *nkx2.1 (RNAi)* worms exhibited a complete loss of all VM GABAergic neurons, while leaving DL neurons intact (*Figure 2B,E*, *Figure 2—figure supplement 1*). Combined with the above dFISH data, the ablation of all VM GABAergic neurons demonstrated that *nkx2.1* acts in the maintenance of this neural subtype.

Interestingly, both *nkx2.1(RNAi)* and *arx(RNAi)* animals exhibited a significant reduction in the number of octopaminergic neurons, which correlated with the degree of expression for each transcription factor within this neural cell type (*Figure 2C,F*). While *nkx2.1(RNAi)* animals exhibited a > 60% reduction in octopaminergic neurons, *arx(RNAi)* worms displayed a more modest but significant ~ 25% reduction (*Figure 2C,F*). It should also be noted that the neurons lost in *nkx2.1 (RNAi)* or *arx(RNAi)* animals were located in the VM region of the planarian brain (*Figure 2C*). As expected, RNAi of *nkx2.1* or *arx* had no appreciable effect on the dopaminergic and serotonergic neuron populations (*Figure 2—figure supplement 2*). Therefore, the maintenance roles of *nkx2.1* and *arx* are limited to the VM cholinergic, GABAergic, and octopaminergic neuronal cell populations.

## *hedgehog* is expressed in VM neural cell types

Previous studies have shown that the planarian CNS is a major site of expression for the single *hedgehog* ligand in planarians (*Smed-hh*; henceforth referred to as *hh*) (*Figure 3A*) (*Rink et al., 2009*; *Yazawa et al., 2009*). However, it is unknown what specific cells express this signal and what cells respond to it. Transverse sections through the brain highlighted *hh* expression within the VM regions of each brain lobe, similar to that of *arx* (*Figure 3B*). More specifically, *hh* expression was observed in the same VM neural cell types as described above for *nkx2.1* and *arx*. As with *arx*, *hh* expression within cholinergic neurons was heavily biased toward the VM region of the brain lobe, where 23.17 ± 2.88% of these neurons expressed the signaling molecule (*Figure 3C*). In addition, *hh* was expressed by nearly all VM GABAergic neurons (98.08 ± 3.85), as well as a considerable proportion of octopaminergic neurons (33.7 ± 2.23%) (*Figure 3D–E*). Although a limited number of DL GABAergic, dopaminergic, and serotonergic neurons exhibited *hh* expression (*Figure 3—figure supplement 1*), VM neurons appear to be the main source of the planarian *hh* signaling molecule.

## *arx(RNAi)* leads to a reduction of *hh* expression from VM neurons

Although only partially overlapping, the finding that *hh* was co-expressed within the same VM neural cell types as *nkx2.1* and *arx* (*Figure 3C–E*) suggested that these two transcription factors may have upstream regulatory function on neuronal *hh* expression. WISH experiments demonstrated a clear reduction in *hh* expression within the medial CNS of *arx(RNAi)* worms (*Figure 3F*; white arrows), while silencing *nkx2.1* had little effect on *hh* expression (*Figure 3—figure supplement 2*). This reduction in *hh* expression was quantified in the same manner described above for cholinergic neurons, by imaging transverse sections through single brain lobes, and counting the number of *hh*[+] cells in the VM and DL brain regions, within 20 μm of confocal depth. This analysis revealed a near 60% reduction in the number of VM *hh*[+] neurons in *arx(RNAi)* animals (*Figure 3F–G*), but no significant loss of *hh*[+] cells in the DL brain (*Figure 3—figure supplement 2*). In contrast, silencing of *nkx2.1* did not significantly alter *hh* expression levels in either brain region (*Figure 3G*, *Figure 3—figure supplement 2*). While it is unclear whether *arx* directly regulates *hh* expression in VM

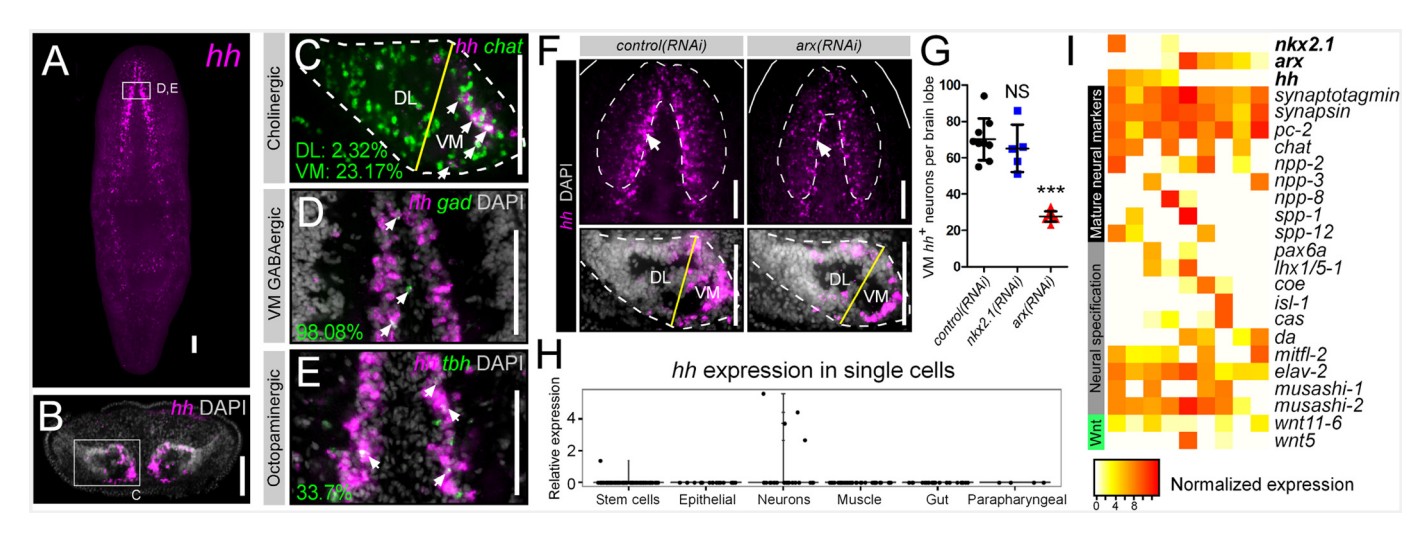

**Figure 3.** *hh* is expressed in VM neural cell types. (**A**) FISH image of whole-mount *hh* expression. (**B**) FISH image of *hh* expression in a brain cross-section (**C–E**) Single confocal plane of dFISH images depicting *hh* expression in cholinergic, GABAergic, and octopaminergic neurons (white arrows). (**F**) Projected confocal image stacks displaying *hh* expression within the brain in RNAi-treated animals. White arrow highlights reduced *hh* expression in the VM region of the brain. Dashed lines represent the outer border of the brain lobe and yellow lines represent the plane of division between the VM and DL brain regions. (**G**) Quantification of VM *hh*+ neurons. Graph is a dot plot, ***p<0.001, error bars are standard deviation. (**H**) Violin plot showing relative expression of *hh* within sequenced single cells across various tissue types. (**I**) Heatmap depicting normalized expression levels of mature neuron markers and genes associated with neuronal specification within nine individually sequenced neurons that also express *arx* or *nkx2.1*. Scale bars = 100 μm. DL, Dorsal-lateral; FISH, Fluorescent in situ hybridization; VM, Ventral-medial.

The following figure supplements are available for figure 3:

**Figure supplement 1.** *hh* is not significantly expressed in DL GABAergic, dopaminergic, or serotonergic neurons.

**Figure supplement 2.** *nkx2.1* is not required for neuronal *hh* expression.

**Figure supplement 3.** *nkx2.1* and *arx* are expressed in *hh*+ cells in the brain

neurons, or whether reduced *hh* expression is related to a loss of VM cholinergic neurons, it is clear that *hh* expression in the VM domain of the planarian brain requires *arx* gene function.

## Single-cell RNA sequencing confirms neuronal expression of *hh*, *nkx2.1*, and *arx*

Recent studies have begun using single-cell RNA sequencing (scRNAseq) technologies, to investigate the complete transcriptional profiles of individual planarian stem cells and mature cell types (*Wurtzel et al., 2015*; *Molinaro and Pearson, 2016*). From the scRNAseq dataset generated by Wurtzel *et al.*, all cells from uninjured animals (including stem cells, neurons, gut, epithelial, muscle, and parapharyngeal cells) as well as neurons from injured worms were analyzed for expression of *hh*. From this group of sequenced cells, *hh* expression was detected in 4/39 neurons, and a single stem cell, but no other cell types (*Figure 3H*), supporting previous data that *hh* expression is largely restricted to the nervous system.

In addition, these 39 sequenced neurons were examined for expression of *nkx2.1* and *arx*, which were detected in two and six neurons, respectively, with two of these cells displaying co-expression with *hh* (*Figure 3I*). Expression of *nkx2.1* or *arx* within *hh*+ neurons in the brain was also directly observed by dFISH (*Figure 3—figure supplement 3*). High expression levels of common mature neuron markers (*Smed-synaptotagmin*, *Smed-synapsin*, *Smed-pc2*) were found in all nine sequenced cells, and more specifically, five of these cells were *chat*+ cholinergic neurons (*Figure 3I*) (*Agata et al., 1998*; *Cebrià, 2008*). In addition, each of these nine neurons expressed a unique

combination of neuropeptides (*Collins et al., 2010*), transcription factors (*Cowles et al., 2013*, *2014*; *Currie and Pearson, 2013*; *Pineda et al., 2002*), RNA-binding proteins (*Higuchi et al., 2008*; *Koushika et al., 1996*), and even 2 Wnts with known or putative roles in neuronal specification and patterning (*Figure 3I*). This in silico data acts to further confirm that the planarian *hh* ligand is expressed by mature neurons.

## Planarian neoblasts expresses *hh* signal transduction machinery

Considering that VM planarian neurons are a major source of the *hh* signaling molecule (*Figure 3*) (*Rink et al., 2009*; *Yazawa et al., 2009*), it was of great interest to discover what cells and tissues might be the target of this brain-derived signaling molecule. Previous observations have shown that *hh* signaling activity can influence global proliferation levels (*Rink et al., 2009*). Therefore, the adult stem cells (ASCs) that occupy the space in between the two brain lobes were examined as a potential target of planarian *hh*.

Reception and effective transduction of the *hh* signaling pathway requires the receptors Patched (*ptc*) and Smoothened (*smo*) as well as the Gli transcription factors (*Ho and Scott, 2002*). Confirming previous reports, WISH experiments for the planarian homologs of these key signal transduction effectors (*Smed-ptc*, *Smed-smo*, and *Smed-gli-1*) (*Rink et al., 2009*) demonstrated relatively ubiquitous expression throughout the body of the adult animal, with *Smed-gli-1* showing a particularly strong signal within the gastrovascular system (*Figure 4A*). However, when *piwi-1*$^+$ ASCs (*Reddien et al., 2005b*) were visualized in between the two brain lobes, many were observed to co-express the *ptc* receptor (*Figure 4B*). Similarly, *smo* and *gli-1* expression was detected in brain-adjacent ASCs (*Figure 4B*), providing support for the notion that planarian stem cells might be a target of brain-derived *hh* signals.

In order to support these dFISH findings, we examined gene expression from a recent dataset (*Molinaro and Pearson, 2016*) containing 96 single stem cells in the head (isolated from a FACS gate termed X1) as well as 72 cells from a FACS gate termed X2 that contains a mixture of stem cells and their immediate division progeny (*Zhu et al., 2015*; *Hayashi et al., 2006*). These cells were examined for expression of *ptc*, *smo*, or *gli-1*. Individual neoblasts were first examined for expression of established stem cell markers and cell cycle regulators (*piwi-1*, *piwi-2*, *piwi-3*, *Smed-bruli*, *Smed-tdrd1L2*, *Smed-chd4*, *Smed-mcm2*, *Smed-pcna*, *Smed-cyclinB-2*) (*Zhu et al., 2015*; *van Wolfswinkel et al., 2014*; *Guo et al., 2006*; *Reddien and Sánchez Alvarado, 2004*). These stem cells were then binned as either X1 or X2 and arranged in descending order based on *ptc* then *smo* then *gli-1* expression (*Figure 4C*). Approximately 30% (56/168 cells) expressed at least one of *ptc*, *smo*, or *gli-1* (*Figure 4C*). Interestingly, approximately half of these sequenced cells (81/168) also expressed at least one of the nine planarian Frizzled (*fz*) receptors (which bind Wnt ligands) (*Gurley et al., 2008*, *2010*). These in silico findings were also confirmed by performing dFISH experiments, which showed expression of all nine *fz* receptors within *piwi-1*$^+$ ASCs in between the two brain lobes (*Figure 4—figure supplement 1*). Parallel expression analyses were performed on another scRNAseq dataset of 84 ASCs from the whole body of the worm with similar conclusions (*Figure 4—figure supplement 1*) (*Wurtzel et al., 2015*). Taken together, these dFISH and in silico data suggested that planarian stem cells adjacent to the CNS are one of the cellular targets of brain-derived *hh* and Wnt signals.

## A subset of *nkx2.1*$^+$ and *arx*$^+$ neoblasts and post-mitotic progenitors exists in close proximity to the VM brain region

Recent studies have highlighted an interesting property of planarian transcription factors: they are expressed not only by the cells and tissues that they act to specify, but also within ASCs nearby the mature organ in question (*Currie and Pearson, 2013*; *Cowles et al., 2013*; *Scimone et al., 2011*, *2014*; *Lapan and Reddien, 2011*, *2012*). Therefore, ASCs nearby the VM brain region were investigated for expression of the *nkx2.1* and *arx* transcription factors. By examining transverse sections through the brain, stem cells, which express *piwi-1* mRNA and PIWI-1 protein, can be observed in great numbers, and are largely excluded from the two brain lobes (*Figure 5A–F*). In addition, their post-mitotic progenitors, which shut down *piwi-1* expression, but briefly maintain detectable amounts of PIWI-1 protein, can be visualized in this region (*Guo et al., 2006*; *Wenemoser and Reddien, 2010*). Some of these progenitors can be found several cell diameters within the mature brain lobes (*Figure 5A–F*; green arrowheads). In this specific spatial region, expression of *nkx2.1* or *arx*

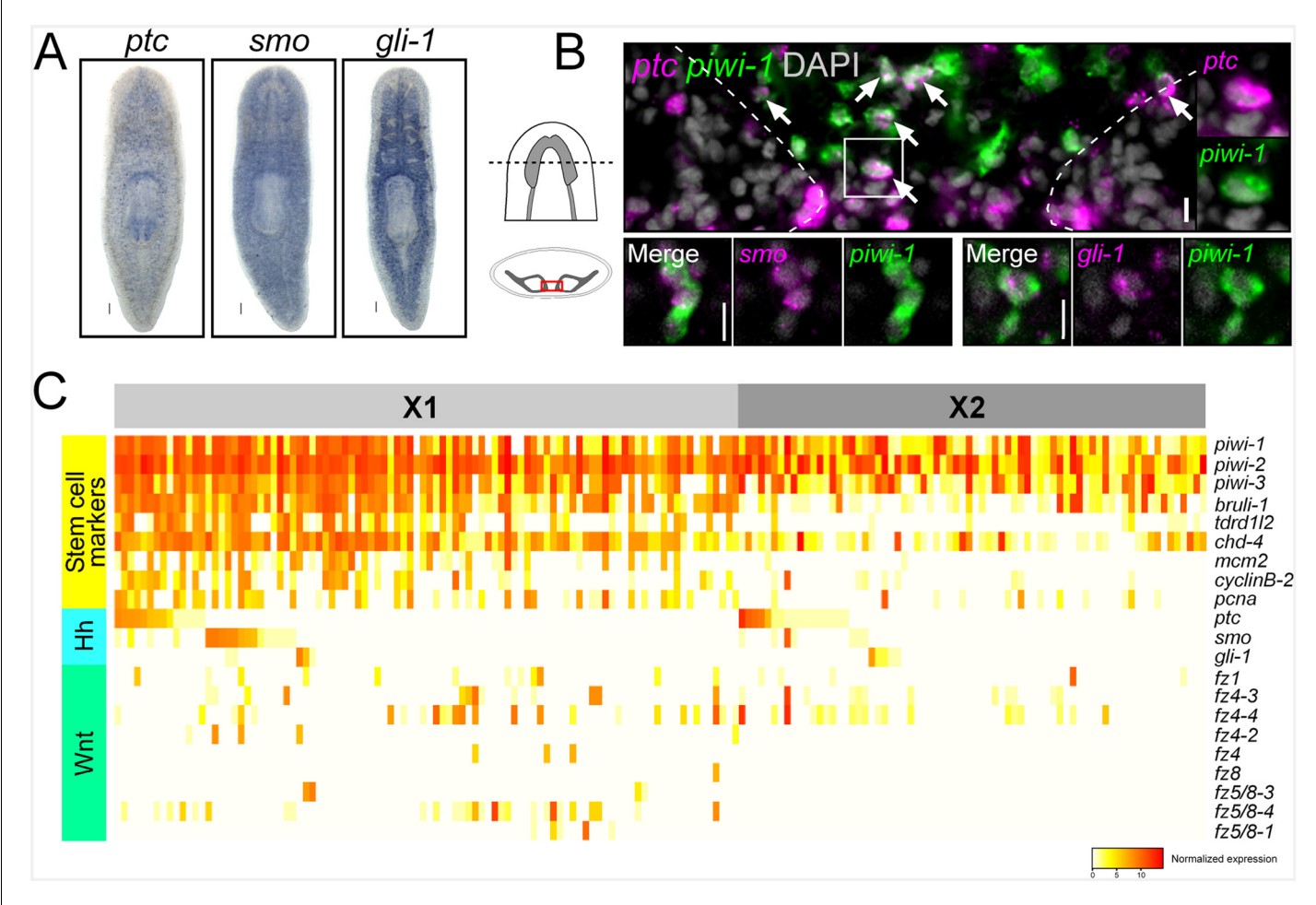

**Figure 4.** Planarian neoblasts express *hh* and Wnt signal transduction effectors. (A) WISH images for *ptc*, *smo*, and *gli-1*. (B) Single confocal planes from dFISH images showing expression of *ptc*, *smo*, and *gli-1* within *piwi-1*⁺ stem cells (white arrows) in between the two brain lobes in cross-section. Dashed lines mark the outer border of the brain lobes. Cell in solid white box is enlarged and channels are split in two right panels. Scale bars for dFISH images = 10 µm. (C) Heatmap depicting normalized expression levels of key *hh* and Wnt signal transduction genes and stem cell markers, within 168 individually sequenced head stem cells (X1) or stem cells + progeny (X2).

The following figure supplement is available for figure 4:

**Figure supplement 1.** Planarian neoblasts express Wnt signal transduction effectors.

was observed within a significant number of stem cells or post-mitotic progenitors (*Figure 5A–F*). Quantification of *piwi-1*⁺ stem cells revealed that a significant percentage displayed expression of *nkx2.1* (7.99 ± 0.95%), whereas *arx* expression was less frequent (1.74 ± 0.99%). Quantification of all PIWI-1⁺ cells (encompassing both stem cells and their post-mitotic progeny) demonstrated an expansion of *nkx2.1* and *arx* expression, to include 13.29 ± 1.96% and 8.11 ± 1.83%, respectively, of all cells counted.

In order to support the expression of *nkx2.1* and *arx* within planarian neoblasts, we again examined the pool of 168×1 and X2 cells from the head of the animal (*Molinaro and Pearson, 2016*). Both *nkx2.1* and *arx* were found to be expressed within these cells and a total of 31/168 cells expressed at least one factor (*Figure 5G*). Importantly, *nkx2.1*⁺ or *arx*⁺ stem cells demonstrated high expression levels of established stem cell markers/regulators (*piwi-1*, *piwi-2*, *piwi-3*, *Smed-bruli*, *Smed-tdrd1L2*, *Smed-chd4*), and cell cycle genes (*Smed-mcm2*, *Smed-pcna*, *Smed-cyclinB-2*) (*Figure 5G*) (*Zhu et al., 2015*; *van Wolfswinkel et al., 2014*; *Guo et al., 2006*; *Reddien and*

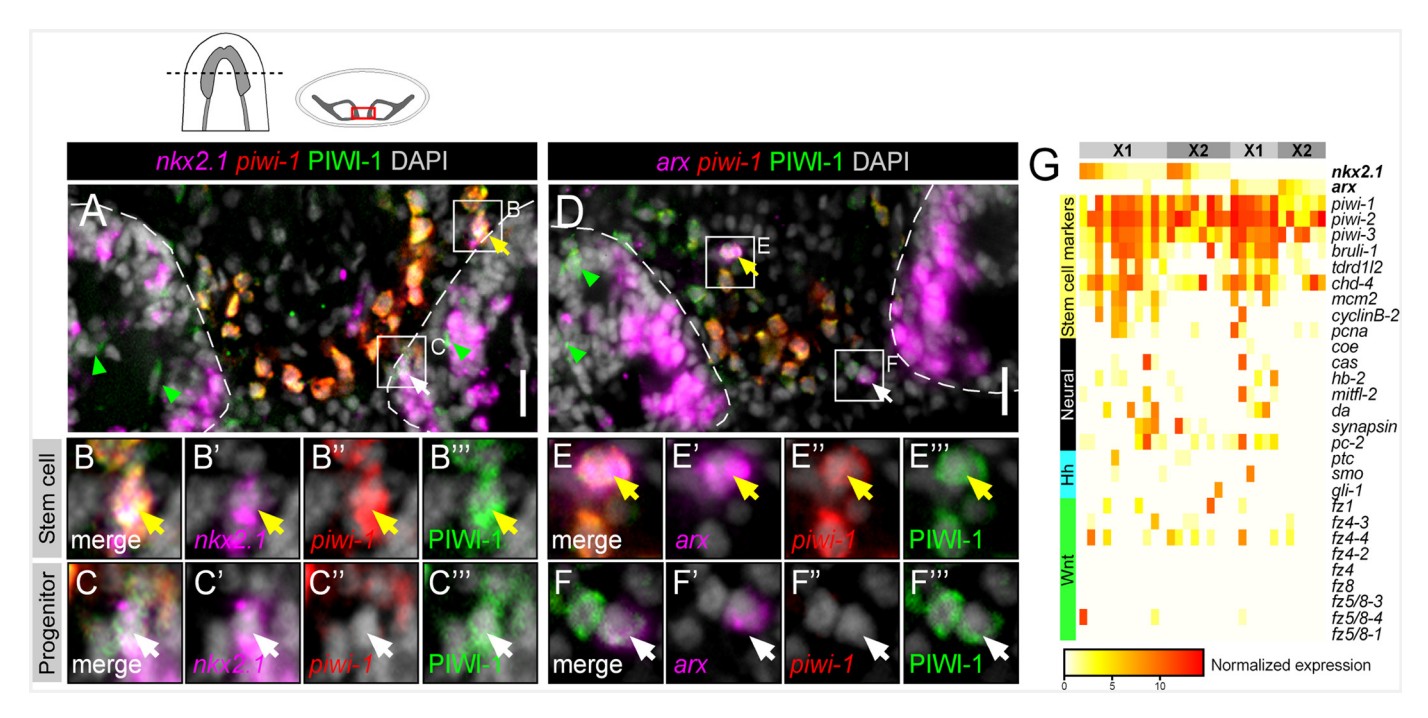

**Figure 5.** *nkx2.1* and *arx* are expressed in stem cells and post-mitotic progenitors. (A–F) Single confocal planes of dFISH images combined with immunolabelling, displaying expression of *nkx2.1* and *arx* within stem cells (yellow arrows) and post-mitotic progenitors (white arrows). Green arrowheads highlight post-mitotic progenitors within the brain lobes. Boxed cells are enlarged below and individual channels are split. Dashed lines mark the outer border of the brain lobes. Cartoon depicts region of interest for all confocal images. Scale bars = 20 µm. (G) Heatmap depicting normalized expression of stem cell markers, genes associated with neurons, *hh*, and Wnt transduction genes, within 31 individually sequenced head stem cells (X1) or stem cells + progeny (X2) that also express *arx* or *nkx2.1*.

The following figure supplement is available for figure 5:

**Figure supplement 1.** *nkx2.1* and *arx* are expressed in *ptc*[+]/PIWI-1[+] cells.

*Sánchez Alvarado, 2004*). In addition, many cells displayed expression of established and putative neuronal transcription factors (*Smed-coe*, *Smed-cas*, *Smed-hb-2*, *Smed-mitfl-2*, *Smed-da*, *Smed-isl-1*) and mature neural markers (*Smed-synapsin*, *Smed-pc-2*) (*Figure 5G*) (*Cowles et al., 2013*, *2014*; *Hayashi et al., 2011*; *Scimone et al., 2014*). Several of these *nkx2.1*[+] or *arx*[+] stem cells also exhibited expression of *ptc*, *smo*, or *gli-1* (7/31 cells) and at least one of the nine *fz* receptors (14/31) (*Figure 5G*). Parallel expression analyses were performed on the dataset of 84 single ASCs from the whole body of the worm with similar conclusions (*Figure 4—figure supplement 1*) (*Wurtzel et al., 2015*). Finally, *ptc* expression within *nkx2.1*[+] and *arx*[+] neural progenitors was also observed in vivo based on dFISH combined with immunostaining against the PIWI-1 protein (*Figure 5—figure supplement 1*). Together, these data demonstrated that putative neural progenitor cells, located adjacent to the mature brain, express the machinery to transduce *hh* signals and may be committed to producing *arx*[+] or *nkx2.1*[+] neurons.

## Hedgehog signaling regulates neural progenitor dynamics

Based on the expression of *hh* within VM neurons and the *hh*-transduction machinery within adult stem cells, the *hh* signaling pathway was next assessed for a potential role in the regulation of neural progenitor cells and the ultimate production of new neurons. This was performed by RNAi knockdown of the *hh* ligand, resulting in decreased signaling activity, or by knockdown of the *ptc* receptor (a negative regulator of pathway activity), resulting in increased *hh* signaling levels (*Rink et al., 2009*). Neural progenitor cells were then visualized by taking transverse sections through the brain, and imaging PIWI-1[+] stem and progenitor cells in between the brain lobes for expression of several

transcription factors of putative neural stem/progenitors (*nkx2.1*, *arx*, *lhx1/5–1*, *coe*, and *pax6a*) (*Figure 6A*) (*Currie and Pearson, 2013*; *Cowles et al., 2014*; *Pineda et al., 2002*; *Scimone et al., 2014*). *hh(RNAi)* and *ptc(RNAi)* animals were investigated for changes to the percentage of PIWI-1$^+$ stem/progenitor cells expressing these neural transcription factors. Silencing of the planarian *hh* ligand resulted in a significant reduction of expression for all five neural transcription factors within PIWI-1$^+$ cells (*Figure 6B*). In contrast, levels of neural progenitor cells were largely unchanged in *ptc (RNAi)* animals, with the exception of *coe* and *pax6a* which exhibited a small but significant increase in PIWI-1$^+$ cells (*Figure 6B*).

The thymidine analog bromodeoxyuridine (BrdU), which is incorporated into dividing stem cells during DNA synthesis and subsequently chases into their post-mitotic progeny, has proven key for planarian lineage-tracing experiments (*Newmark and Sánchez Alvarado, 2000*), some of which

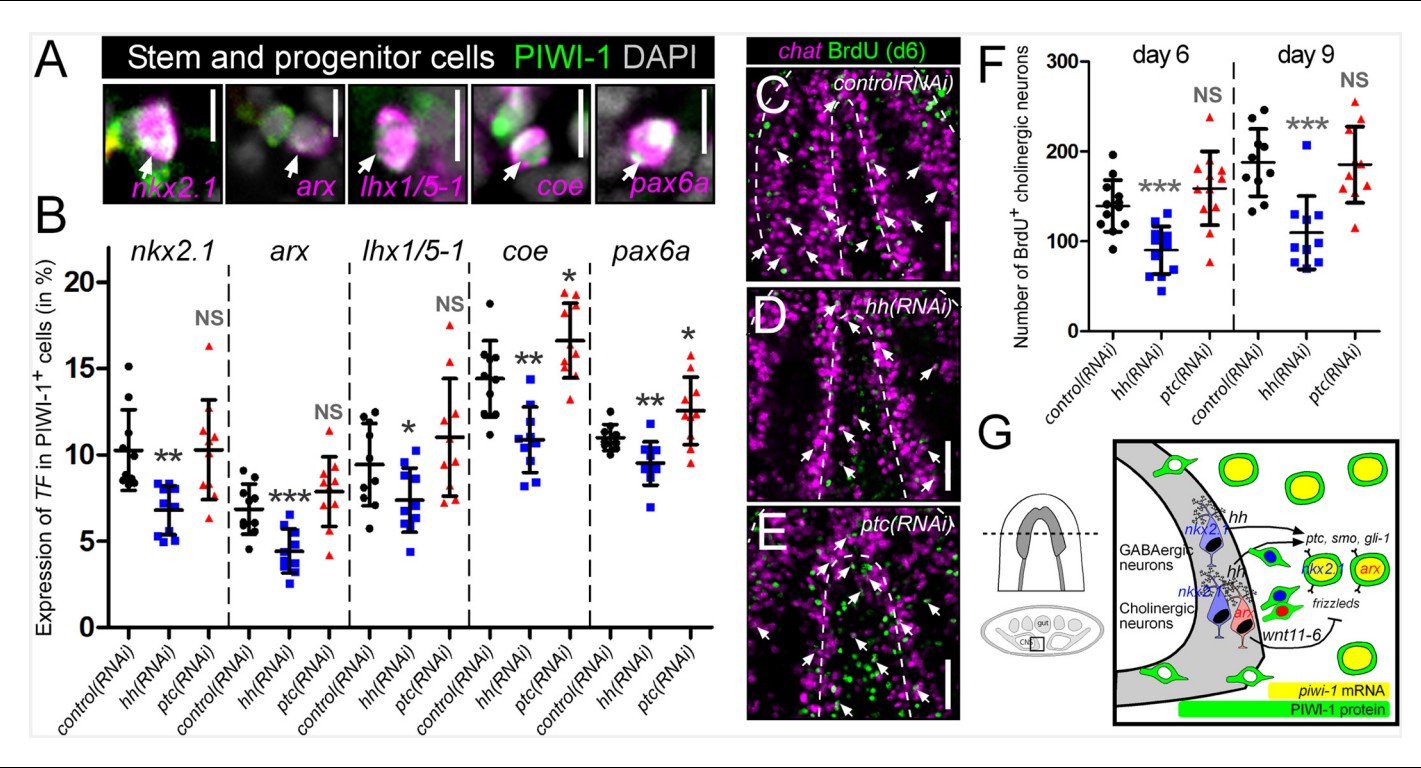

Figure 6. *hh* signaling regulates homeostatic neurogenesis. (A) Single confocal plane images displaying examples of neural progenitor cells (white arrow), PIWI-1$^+$ stem/progenitor cells that express neural transcription factors. Scale bar = 10 µM. (B) Quantification of neural progenitor levels in animals with altered levels of *hh* signaling. Graphs are dot plots measuring the percentage of PIWI-1$^+$ cells that express each neural transcription factor. n = 10, *p<0.05, **p<0.01, ***p<0.001, error bars are standard deviation. (C–E) Single confocal planes displaying newly generated cholinergic neurons in the planarian brain 6 days after a single BrdU pulse (white arrows). Scale bar = 100 µm. (F) Dot plot quantifying the number of new cholinergic neurons. n ≥ 10, ***p<0.001, error bars are standard deviation. (G) Model displaying *nkx2.1*$^+$ and *arx*$^+$ neural stem and progenitor cells and the mature neurons that they produce. These same neural cell types also express the *hh* signaling molecule, which signals back onto adult stem cells, maintaining normal proliferation levels, and homeostatic neurogenesis.

The following figure supplements are available for figure 6:

**Figure supplement 1.** *hh* signaling levels do not affect existing neuronal subpopulations in the intact brain.

**Figure supplement 2.** *hh* signaling levels do not affect existing neuronal subpopulations in the intact brain Projected confocal images showing dopaminergic and serotonergic neurons after six RNAi feedings to knockdown *hh* or *ptc*.

**Figure supplement 3.** *hh* signaling levels do not affect other planarian progenitor populations.

**Figure supplement 4.** *hh* signaling levels do not affect BrdU incorporation into other planarian tissues.

have revealed that the intact planarian CNS maintains high levels of continuous neurogenesis (*van Wolfswinkel et al., 2014*; *Zhu et al., 2015*). Therefore, in order to investigate whether *hh* signaling levels regulate neuronal homeostasis and the production of new cholinergic neurons, a BrdU pulse-time chase experiment was performed. Following gene silencing of *hh* or *ptc*, RNAi-treated animals were exposed to a single pulse of BrdU. After 6- and 9-day chase periods, newly generated cholinergic neurons were detected by FISH for *chat* expression as well as nuclear BrdU staining (*Figure 6C–E*). The level of neurogenesis was then quantified by counting the number of *chat*+BrdU+ cells observed within 30-µm-thick confocal stacks from within the planarian brain. *hh(RNAi)* animals consistently displayed significantly lower cholinergic neural homeostasis both after 6-day (~35% reduction) and 9-day chase periods (>40% reduction), while silencing of *ptc* had little effect (*Figure 6C–F*). To investigate whether this decrease in neuronal homeostasis levels affected the intact nervous system, long-term RNAi experiments were performed (10 feedings), and all five neuronal populations were examined. Of particular interest were the VM cholinergic, GABAergic, and octopaminergic neural populations, which express the *hh* ligand (*Figure 3C–E*). However, decreased *hh* signaling did not affect the intact population of any neuronal cell type (*Figure 6—figure supplements 1* and *2*; see Discussion).

To determine whether brain-derived *hh* signals were required specifically to maintain normal levels of neurogenesis, or whether this ligand acts globally to stimulate the production of all planarian cell types, progenitor cells and BrdU incorporation were examined for four additional tissues; the eyes, pharynx, epidermis, and gut. Importantly, these other planarian tissues were largely unaffected by perturbations to *hh* signaling, except for the eyes, which exhibited reduced progenitor cell numbers in *hh(RNAi)* animals, and increased BrdU incorporation after *ptc* knockdown (*Figure 6—figure supplements 3* and *4*). These combined experiments, demonstrated that neuronal sources of the planarian *hh* ligand specifically regulate their own homeostasis through distinct progenitors in the uninjured brain.

## Discussion

Planarians not only constantly turnover their CNS at relatively high rates, but must also be able to respond to massive injury to regenerate an entire brain de novo. This work has demonstrated that two planarian homeodomain transcription factor homologs, *nkx2.1* and *arx*, function in the long-term maintenance of cholinergic, GABAergic, and octopaminergic neurons in the VM region of the planarian CNS (*Figure 2*). Surprisingly, these same three VM neural cell types were also found to express the *hh* signaling molecule (*Figure 3C–E*). Neuronal *hh* expression was also confirmed by using a scRNAseq dataset that included mature neurons (*Wurtzel et al., 2015*), and additionally revealed that all four of the sequenced *hh*+ neurons co-expressed a planarian Wnt ligand, *Smed-wnt11-6* (*Figure 3I*). Specialized *nkx2.1*+ or *arx*+ neoblasts were also identified, which were typically localized adjacent to the VM region of the planarian CNS, and express key members of the *hh* and Wnt signal transduction machinery (*Figure 5*). Finally, the planarian *hh* pathway was found to be required to maintain steady levels of neurogenesis in the intact brain, as a reduction in *hh* signaling activity, led to decreased production of neural stem/progenitor cells and newly generated cholinergic neurons (*Figure 6G*).

Similar to planarian muscle cells, many of which express important signaling molecules that maintain positional control along the body axis and during regeneration events (*Witchley et al., 2013*; *Scimone et al., 2016*), mature planarian neurons express *hh* and Wnt ligands which regulate neurogenesis levels and maintain patterning (*Hill and Petersen, 2015*). Previous studies have shown that the Wnt ligand *Smed-wnt11-6* and Wnt antagonist *Smed-notum* are expressed by cholinergic neurons, and that this Wnt signal acts on planarian stem cells to repress neuron production and prevent posterior expansion of brain tissue during regeneration (*Hill and Petersen, 2015*). Interestingly, we show that several planarian neurons simultaneously express this repressive Wnt ligand as well as the *hh* signaling molecule, which promotes the production of neurons (*Figures 3I* and *6G*). Conceptually, it makes sense that signals originating from the mature CNS, would act to repress neurogenesis programs. This would help to establish a steady-state system, which modulates neurogenesis levels depending on the size of the existing CNS. However, our observations suggest that contrary to this model, neuronal sources of *hh* actively promote neurogenesis levels in both the brain and of eye photoreceptors neurons (*Figure 6—figure supplements 3* and *4*).

It is unclear, exactly how the planarian *hh* signaling pathway achieves rates of neuronal homeostasis, whether it acts as a permissive cue, influencing neoblasts adjacent to the brain to produce neural cell types, or as a mitogenic signal, to control proliferation levels of neural-biased stem cells. Similar to mechanisms of Hh action in other systems, the regulatory loops are likely to be complex in planarians due to the fact that long-term *hh(RNAi)*, surprisingly, did not produce robust neural deficits over a span of five weeks (*Figure 6—figure supplements 1* and *2*). Perhaps with body-wide changes in proliferation in *hh(RNAi)*, rates of neural cell death are also decreased, leading to little change in neuronal populations (*Rink et al., 2009*). Similarly, in the mammalian CNS, expression of Shh from differentiating and mature neurons is required to maintain normal proliferation levels of embryonic neural precursors and adult neural stem cells, respectively (*Álvarez-Buylla and Ihrie, 2014*; *Ihrie et al., 2011*; *Fuccillo et al., 2006*).

Unlike most organisms where neural stem cells are packed within an organized neuroepithelium, planarian neural-biased stem cells are situated in the mesenchymal space in between the two brain lobes (*Figure 5*). While this loose grouping of stem cells offers few similarities to a true neuroepithelium in terms of structure or cellular origin, the local signaling microenvironment may fulfill a similar function. Computationally, neural-committed stem cells, termed vNeoblasts, were recently detected and may be the targets of brain-derived signals (*Molinaro and Pearson, 2016*). However, while $nkx2.1^+$ and $arx^+$ stem cells exhibit a spatial distribution and transcriptional profile suggestive of neuronal lineage commitment (*Figure 5G*), without definitive lineage-tracing studies, these neoblasts cannot yet be classified as true neural stem cells.

In chordate nerve cord development, Hh is expressed ventrally, along the long body axis and has strong ventralizing roles in cell fate determination (*Briscoe and Ericson, 2001*; *Jessell, 2000*). This is not typical of arthropods (*Matise, 2007*; *Arendt and Nübler-Jung, 1999*). It is interesting to speculate that VM neural fates in planarians are specified using mechanisms more associated with chordates. In support of this idea, the role of *nkx2.1* and *arx* in the maintenance of planarian cholinergic and GABAergic neurons was particularly noteworthy (*Figure 2*), as their mammalian counterparts (Nkx2.1 and Arx) are known to fulfill similar roles and also be under the control of *sonic hedgehog* (*Jessell, 2000*; *Colasante et al., 2008*). In the embryonic rodent ventral telencephalon, Nkx2.1 acts in upstream neural progenitor cells to produce both cortical GABAergic interneurons and striatal cholinergic interneurons (*Butt et al., 2008*; *Lopes et al., 2012*; *Sussel et al., 1999*), whereas Arx functions downstream in the terminal differentiation and migration of cortical GABAergic interneurons (*Vogt et al., 2014*). Notwithstanding this slight deviation in cell fate determinism, it appears that these two transcription factors have retained a remarkable degree of functional conservation across this significant evolutionary gap.

## Materials and methods

### Cloning

*Smed-nkx2.1* and *Smed-arx* were found by homology to mammalian orthologs in the sequenced and assembled planarian genome and transcriptomes as previously described (*Currie and Pearson, 2013*; *Robb et al., 2008*). Primers were designed and full length genes were cloned by 3' RACE. *Smed-nkx2.1* is transcript number dd_Smed_v6_13898_0_1 from (http://planmine.mpi-cbg.de/, *Brandl et al., 2016*) and *Smed-arx* was deposited to Genbank under the accession number KX961610.

### Animal husbandry and RNAi

Asexual *S. mediterranea* CIW4 strain were reared as previously described (*Sánchez Alvarado et al., 2002*). RNAi experiments were performed using previously described expression constructs and HT115 bacteria (*Newmark et al., 2003*). Briefly, bacteria were grown to an O.D.600 of 0.8 and induced with 1 mM IPTG for 2 hr. Bacteria were pelleted and mixed with liver paste at a ratio of 500 µl of liver per 100 ml of original culture volume. Bacterial pellets were thoroughly mixed into the liver paste and frozen as aliquots. The negative control, '*control(RNAi)*', was the *unc22* sequence from *C. elegans* as previously described (*Reddien et al., 2005a*). All RNAi food was fed to 7-day starved experimental worms every third day for six total feedings, and fixed 10–14 after the final RNAi

feeding. All animals used for immunostaining were 2–3 mm in length and size-matched between experimental and control worms.

## Immunolabeling, in situ hybridizations (ISH), and BrdU

WISH, dFISH, and immunostaining were performed as previously described (*Currie et al., 2016*). Colorimetric WISH stains were imaged on a Leica M165 fluorescent dissecting microscope. dFISH and fluorescent mouse-anti-PIWI-1 (gift from Dr. Jochen Rink) stains were imaged on a Leica DMIRE2 inverted fluorescence microscope with a Hamamatsu Back-Thinned EM-CCD camera and spinning disc confocal scan head. Cell counts and co-localizations were quantified using freely available ImageJ software (http://rsb.info.nih.gov/ij/). Significance was determined by a two-tailed Student's *t*-test. All experiments were, at minimum, triplicated and at least five worms were used per stain and per time point. All labeling images were post-processed using Adobe Photoshop.

## Analysis of scRNAseq data

Raw scRNAseq data from head-specific X1 and X2 cells from (*Molinaro and Pearson, 2016*) can be downloaded under NCBI Gene Expression Omnibus (GEO accession GSE79866). Raw scRNAseq data from uninjured cells (including stem cells, neurons, gut, epithelial, muscle and parapharyngeal cells) as well as neurons following injury generated by Wurtzel *et al.*, were obtained from the NCBI Sequence Read Archive (SRA:PRJNA276084). Reads were aligned to the SmedASXL transcriptome assembly under NCBI BioProject PRJNA215411 using bowtie2 (*Langmead and Salzberg, 2012*) with 15 bp 3' trimming. The read count data were log2-transformed (log2(count + 1)), violin plot was produced using modified source code from (*Macosko et al., 2015*) and the heatmaps were produced using the modified heatmap.3 source code from (*Molinaro and Pearson, 2016*).

## Acknowledgements

KWC and AMM were funded by Natural Sciences and Engineering Research Council of Canada (NSERC) grant # RGPIN-2016–06354. BJP was funded by Ontario Institute for Cancer Research (OICR) grant number IA-026. We thank Dr. Ricardo Zayas for comments on the manuscript.

## Additional information

### Funding

| Funder | Grant reference number | Author |
|---|---|---|
| Natural Sciences and Engineering Research Council of Canada | RGPIN-2016-06354 | Ko W Currie<br>Alyssa M Molinaro |
| Ontario Institute for Cancer Research | IA-026 | Bret J Pearson |

The funders had no role in study design, data collection and interpretation, or the decision to submit the work for publication.

### Author contributions

KWC, Conception and design, Acquisition of data, Analysis and interpretation of data, Drafting or revising the article; AMM, Conception and design, Analysis and interpretation of data; BJP, Conception and design, Analysis and interpretation of data, Drafting or revising the article

### Author ORCIDs

Bret J Pearson, http://orcid.org/0000-0002-3473-901X

## Additional files

### Major datasets

The following previously published datasets were used:

| Author(s) | Year | Dataset title | Dataset URL | Database, license, and accessibility information |
|---|---|---|---|---|
| Wurtzel O, Cote LE, Poirier A, Satija R, Regev A, Peter W | 2015 | Schmidtea mediterranea Transcriptome or Gene expression | https://www.ncbi.nlm.nih.gov/bioproject/276084 | Publicly available at the NCBI BioProject database (accession no: PRJNA276084) |
| Molinaro AM, Pearson BJ | 2016 | In silico lineage tracing through single cell transcriptomes identifies a neural stem cell population in planarians | https://www.ncbi.nlm.nih.gov/geo/query/acc.cgi?acc=GSE79866 | Publicly available at the NCBI Gene Expression Omnibus (accession no: GSE79866) |

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
