## [Decision Letter]

Thank you for submitting your article "Neuronal sources of *hedgehog* modulate neurogenesis in the adult planarian brain" for consideration by *eLife*. Your article has been favorably evaluated by Marianne Bronner (Senior Editor) and three reviewers, one of whom, Yukiko M Yamashita (Reviewer #1), is a member of our Board of Reviewing Editors.

The reviewers have discussed the reviews with one another and the Reviewing Editor has drafted this decision to help you prepare a revised submission.

Review summary:

Adult neurogenesis in the brain is an intriguing phenomenon that is widespread among animal systems, but its regulation is currently not well understood. Planarians are an accessible model with a high rate of adult neuronal cell turnover, and therefore are an excellent model to study the regulation adult neurogenesis in vivo. This manuscript addresses the contribution of hedgehog signaling to the regulation of several neuronal cell populations in the planarian brain. The text is well written, the work is elegant and well executed, and the images look clear. Its findings should be of interest to a wide audience interested in neurogenesis.

These authors investigate mechanisms controlling neurogenesis in regeneration and characterize two transcription factors (*arx* and *nkx2.1*) important for neural differentiation in the planarian brain. *nkx2.1* was expressed in both octopaminergic and GABAergic neurons whereas *arx* was expressed mainly in octopaminergic neurons within a ventromedial (VM) region. *nkx2.1* RNAi eliminated VM GABAergic neurons and strongly reduced numbers of octopaminergic neurons. *arx* RNAi animals had reduced numbers of VM-resident cholinergic and octopaminergic neurons. Searching for signals originating in this ventromedial brain domain that could regulate neurogenesis, they find that hedgehog is expressed within VM neurons. The authors analyze a prior single-cell RNAseq study to identify *hedgehog^+^* cells as being mainly neuronal, with the feedback target patched expressed in several populations of neoblasts including brain progenitors. Furthermore, *hh* RNAi animals had reduced numbers of brain progenitors within the neoblast population, including *nkx2.1*^+^ and *arx*^+^ neoblasts, and also these animals had reduced incorporation of new *chat*^+^ neurons into the brain, arguing for a direct function for *hh* signaling in promoting neurogenesis. Finally, this differentiation program has importance for CNS function as *nkx2.1* and *arx* inhibited animals present with behavioral abnormalities.

As you can see in the specific comments provided below, the reviewers agreed that there are a few gaps in the manuscript. In particular, the following points must be thoroughly addressed, which the reviewers agreed to be straightforward. Thus, we would like to invite you to submit a revised version of the manuscript.

Essential revisions:

1) The conflicting observations in the *hh* RNAi phenotype must be resolved: increased BrdU labeling of neuronal lineages combined with the unaltered levels of mature neurons.

This could and should be addressed by:

As confirmed by the authors, *hh* is known to increase cell proliferation in general. This notion is supported by the fact that most neoblasts express the machinery required to respond to *hh* signalling. Therefore, BrdU incorporation in the brain should be normalised to incorporation in other tissues to show that the effect is real and is specific to neurons. (This would not require new experiments, as this data can probably be extracted from the same slides they used before.)

As a possible alternative if normalization in this way proves to be difficult, I think they could address this issue by quantifying progenitors from some unrelated lineages to determine whether the *hh* RNAi affects are specific to neural progenitors or affect other progenitors in the animal more broadly. If the effect of *hh* turns out to be broad then the model would have to be rewritten to incorporate this information, in particular because the positive feedback mechanism would be rather indirect.

Writing an explanation if the conflicting data persist (for example, if the conflicting data are real, the phenotype could arise from alterations in both the rate of apoptosis and the rate of cell differentiation in neurons in response to *hh* signaling. Solving this does require new experiments, and it would be desirable, but possibly not vital. In that case, though the claim of the paper would probably have to be tempered, because the effect of *hh* would be not as well defined.)

2) There is no solid co-expression data for *arx/nkx2.1* and *ptc*, as well as for *arx /nkx2.1* and *hh*. As this is what the paper hinges on, it should be addressed.

3) The Wnt pathway data seem unconnected to the message of the paper, and that removing that data would improve the storyline. If the authors have a compelling reason to keep it in, they may be able to write that more clearly.

[Editors' note: further revisions were requested prior to acceptance, as described below.]

Thank you for submitting your article "Neuronal sources of *hedgehog* modulate neurogenesis in the adult planarian brain" for consideration by *eLife*. Your article has been reviewed by two peer reviewers, and the evaluation has been overseen by a Reviewing Editor and Marianne Bronner as the Senior Editor. The reviewers have opted to remain anonymous.

The reviewers have discussed the reviews with one another and the Reviewing Editor has drafted this decision to help you prepare a revised submission.

As you can see in the reviewers' comments, they have agreed that you have mostly addressed the concerns raised during the first round of the review. However, reviewer #3 pointed out, and the others agreed, that the choice of dataset does not seem ideal, and we would like you to provide explanation for this. Reviewer #3 also raised a few other points. Please revise your manuscript accordingly, and we will be most likely ready to accept your manuscript at that point.

We're looking forward to receiving your revised manuscript.

*Reviewer #2:*

In this revised manuscript, the authors strengthen the model that *hedgehog* signals from the ventromedial brain region positively influence neoblast differentiation into brain cells. New experiments using double FISH provide convincing evidence to confirm the RNA-seq observations of co-expression of *nkx2.1* and *arx* in *hh^+^* cells. The authors also now convincingly demonstrate that effects of *hh* RNAi and *ptc* RNAi on BrdU incorporation and progenitor numbers is specific to the CNS (and eye) and not pharynx, epidermis and gut. Therefore, the influence of *hh* and *ptc* on brain progenitors are not likely to be due to a global alteration of neoblast proliferation or differentiation rates. Finally, text changes have improved the flow of the paper, including the change to the discussion/presentation of the Wnt data which has improved the focus of the paper. Therefore, I believe the authors have satisfactorily addressed the issues initially raised, and the manuscript is ready for publication.

*Reviewer #3:*

I think most issues raised in the decision letter have been addressed and this has definitely improved the manuscript. The manuscript is fine, but unfortunately the authors still leave me with the impression that something is being held back. That could be just protecting the start of another paper, or it could be something more. It just keeps me less enthusiastic than I would wish to be.

Specific comments:

The single cell data remains unconvincing as before as it involves too few relevant cells. In addition, when looking into the now declared *nkx2.1* contig in the Wurtzel data its expression seems to be all over the place, and not enriched in neuronal cells. *Arx* seems hardly detectable in the dataset. This is not a fault in the dataset, but it is just not intended for this type of detailed analysis scrutinising expression data of single genes in such low numbers of cells. It is puzzling that the author's group has recently published a much more appropriate single cell dataset, focussing on dividing cells from the head region, yet they have chosen not to strengthen their case with that data.

For the mature neurons, the newly added double FISH clearly shows that there is some overlap between *arx* and *hh*, but also that both are largely non-overlapping in the same region. *nkx2.1* only shows overlap when *hh* signal is oversaturated in the image. Therefore, it is correct that the VM region is the source of *hh* (but it is not shown whether the *arx/nkx2.1* cells are the main source – although that is suggested).

Subsection “*arx* regulates the fate of *hh*^+^ neurons”: title is overstated.*arx* RNAi leads to reduction of *hh* expression from VM neurons. It is not clear whether it has an effect on neuronal fate – let alone regulates that fate.

Subsection “*arx* regulates the fate of *hh^+^* neurons”, first paragraph: Idem. *hh* is expressed in VM neural cells, and *nkx-2.1* and *arx* are, but they are only partially overlapping.

Subsection “Planarian neoblasts expresses *hh* signal transduction machinery”, last sentence: Conclusion is overstated. I don't see how the data suggests that stem cells next to the CNS are the major targets of *hh* signalling. It looks like ~half the zeta cells and ~half the sigma cells have *ptc*. There is no clear data showing that these cells are enriched close to the CNS as the scRNAseq data is not uniquely from head regions. In addition, the zeta cells are very unlikely to be neuronal progenitors, so this rather suggests that it is not a functional relationship.

---

## [Author Response]

*[…] As you can see in the specific comments provided below, the reviewers agreed that there are a few gaps in the manuscript. In particular, the following points must be thoroughly addressed, which the reviewers agreed to be straightforward. Thus, we would like to invite you to submit a revised version of the manuscript.*

*Essential revisions:*

1) The conflicting observations in the hh RNAi phenotype must be resolved: increased BrdU labeling of neuronal lineages combined with the unaltered levels of mature neurons.

*This could and should be addressed by:*

*As confirmed by the authors, hh is known to increase cell proliferation in general. This notion is supported by the fact that most neoblasts express the machinery required to respond to hh signalling. Therefore, BrdU incorporation in the brain should be normalised to incorporation in other tissues to show that the effect is real and is specific to neurons. (This would not require new experiments, as this data can probably be extracted from the same slides they used before.)*

As a possible alternative if normalization in this way proves to be difficult, I think they could address this issue by quantifying progenitors from some unrelated lineages to determine whether the hh RNAi affects are specific to neural progenitors or affect other progenitors in the animal more broadly. If the effect of hh turns out to be broad then the model would have to be rewritten to incorporate this information, in particular because the positive feedback mechanism would be rather indirect.

*Writing an explanation if the conflicting data persist (for example, if the conflicting data are real, the phenotype could arise from alterations in both the rate of apoptosis and the rate of cell differentiation in neurons in response to hh signaling. Solving this does require new experiments, and it would be desirable, but possibly not vital. In that case, though the claim of the paper would probably have to be tempered, because the effect of hh would be not as well defined.)*

These were excellent revision comments, and we have taken extensive steps to resolve these issues. Specifically, we have looked at any possible changes after *hh* or *ptc* RNAi to progenitor levels for other planarian tissue types (eyes, pharynx, epidermis, and gut). In addition, we performed new experiments to examine BrdU incorporation into these same mature tissues after *hh* or *ptc* RNAi. We have found that these RNAi conditions do not reduce the production of progenitors nor BrdU incorporation for the pharynx, epidermis, or gut (Figure 6—figure supplement 3 and Figure 6—figure supplement 4). However, for the eyes, we do observe a decrease in the production of eye progenitors after *hh* RNAi, and an increase in BrdU^+^ cells in the mature eye after *ptc* RNAi. It should be noted that the marker used to identify the eyes, *Smed-ovo*, largely marks the photoreceptor neurons of the visual system (Lapan and Reddien, 2012). Therefore, it appears that *hh* signaling also affects the production of neurons of the visual system in addition to modulating neurogenesis in the CNS. This is now addressed in the Discussion as well.

*2) There is no solid co-expression data for arx/nkx2.1 and ptc, as well as for arx /nkx2.1 and hh. As this is what the paper hinges on, it should be addressed.*

Another good point. We have addressed these points, by performing dFISH experiments to show co-expression of *nkx2.1* and *arx* within *hh*^+^ neurons in the CNS (Figure 3—figure supplement 3). As well we have performed dFISH combined with immunolabelling against a PIWI-1 antibody (which labels stem cells and immediate post-mitotic progenitors), to observe *nkx2.1* and *arx* expression within *ptc*^+^/PIWI-1^+^ cells in between the brain lobes (Figure 5—figure supplement 1). These two points have been incorporated into the re-submission.

*3) The Wnt pathway data seem unconnected to the message of the paper, and that removing that data would improve the storyline. If the authors have a compelling reason to keep it in, they may be able to write that more clearly.*

We agree that the Wnt signaling components may be interfering with the overall clarity of the manuscript. As such, we have dropped a lot of this data from the main figures and the text. Specifically, we have removed all of the WISH and dFISH images for the planarian Frizzled receptors from Figure 4, but have kept them as a supplemental figure, and have significantly cut down their emphasis in the manuscript text. However, we have kept the in silico sequencing data showing that planarian neurons can co-express both the *hh* ligand as well as the *wnt5* and *wnt11-6* ligands, as this is an interesting biological phenomenon where mature neurons express multiple important signaling molecules.

[Editors' note: further revisions were requested prior to acceptance, as described below.]

*[…] As you can see in the reviewers' comments, they have agreed that you have mostly addressed the concerns raised during the first round of the review. However, reviewer #3 pointed out, and the others agreed, that the choice of dataset does not seem ideal, and we would like you to provide explanation for this. Reviewer #3 also raised a few other points. Please revise your manuscript accordingly, and we will be most likely ready to accept your manuscript at that point.*

*We're looking forward to receiving your revised manuscript.*

*Reviewer #3:*

*I think most issues raised in the decision letter have been addressed and this has definitely improved the manuscript. The manuscript is fine, but unfortunately the authors still leave me with the impression that something is being held back. That could be just protecting the start of another paper, or it could be something more. It just keeps me less enthusiastic than I would wish to be.*

*Specific comments:*

*The single cell data remains unconvincing as before as it involves too few relevant cells. In addition, when looking into the now declared nkx2.1 contig in the Wurtzel data its expression seems to be all over the place, and not enriched in neuronal cells. Arx seems hardly detectable in the dataset. This is not a fault in the dataset, but it is just not intended for this type of detailed analysis scrutinising expression data of single genes in such low numbers of cells. It is puzzling that the author's group has recently published a much more appropriate single cell dataset, focussing on dividing cells from the head region, yet they have chosen not to strengthen their case with that data.*

The Wurtzel data are the only in existence to include planarian neurons, so they must be used for the neuronal analysis for the heatmap in Figure 3. Even with only 39 neurons, *arx^+^hh^+^* and *nkx2.1+hh*^+^ cells are detected. The reviewer’s comment regarding broad *nkx2.1* detection in the Wurtzel data simply describes its broad expression pattern shown in Figure 1 and it is not relevant whether it is neuronally “enriched”. It is also not surprising to detect few *arx*^+^ neurons given the very narrow expression of this gene shown in Figure 1.

Our recently published stem cell and progenitor, single-cell sequencing from heads (~200 cells total) could have been used on the stem cell side of *arx* and *nkx2.1* expression, but it did not make sense to us as to why we now chose to use a different dataset for stem cells than we used for neurons (where we had no choice). In addition, using data generated by a different group seemed more robust. However, now based on this reviewer’s comment, we have switched out the Wurtzel data for our own data that was exclusively head stem/progenitors (Molinaro and Pearson, 2016). As expected, we detect many more *nkx2.1*^+^ or *arx*^+^ stem/progenitor cells, supporting all of our same conclusions at the Wurtzel data. The accompanying text now reflects not needing to describe σ, zeta, or γ neoblasts (from old Figure 4), but now must introduce X1/Χ2 terminology where appropriate. The Wurtzel heatmaps from our previous version are still included but now in Figure 4—figure supplement 1.

We strongly disagree with the assessment that these datasets are “not intended for this type of detailed analysis” and actually argue the opposite.

*For the mature neurons, the newly added double FISH clearly shows that there is some overlap between arx and hh, but also that both are largely non-overlapping in the same region. nkx2.1 only shows overlap when hh signal is oversaturated in the image. Therefore, it is correct that the VM region is the source of hh (but it is not shown whether the arx/nkx2.1 cells are the main source – although that is suggested).*

One issue raised by the reviewer above is the relatively low expression of *arx* and *nkx2.1* in scRNAseq data. This also translates to dFISH experiments, where *nkx2.1* and *arx* are technically challenging to detect with standard tyramide amplification. The same is true of *hh* by sequencing and FISH. Therefore, we know that both the sequencing and dFISH are an underestimation of the overlap of these various gene expression patterns due to being near the limits of detection for both methods. We did not, of course, expect much overlap between *nkx2.1* and *hh* in *chat*^+^ neurons due to only ~36% of VM *chat*+ cells express *nkx2.1* and only ~23% express *hh*, in addition to the lack of phenotype in *nkx2.1(RNAi)* on *hh^+^* neurons (Figure 3). We do expect *nkx2.1* and *hh* expression to overlap in GABAergic and Octopaminergic neurons (Figure 1; Figure 3; Figure 3—figure supplement 1). We have now quantified these stains and shown that 63/353 *hh^+^* cells also express *nkx2.1* (Figure 3—figure supplement 3). *arx*, on the other hand, has much more overlap with *hh* expression (191/269 neurons counted), and this is no surprise given the strong phenotype of *arx(RNAi)* (Figure 3).

*Subsection “arx regulates the fate of hh^+^ neurons”: title is overstated.arx RNAi leads to reduction of hh expression from VM neurons. It is not clear whether it has an effect on neuronal fate – let alone regulates that fate.*

Agreed. We have changed the section title to what you suggest.

*Subsection “arx regulates the fate of hh^+^ neurons”, first paragraph: Idem. hh is expressed in VM neural cells, and nkx-2.1 and arx are, but they are only partially overlapping.*

This sentence now reads: “Although only partially overlapping, the finding that *hh* was co-expressed within the same VM neural cell types as *nkx2.1* and *arx* (Figure 3) suggested that these two transcription factors may have upstream regulatory function on neuronal *hh* expression.”

*Subsection “Planarian neoblasts expresses hh signal transduction machinery”, last sentence: Conclusion is overstated. I don't see how the data suggests that stem cells next to the CNS are the major targets of hh signalling. It looks like ~half the zeta cells and ~half the sigma cells have ptc. There is no clear data showing that these cells are enriched close to the CNS as the scRNAseq data is not uniquely from head regions. In addition, the zeta cells are very unlikely to be neuronal progenitors, so this rather suggests that it is not a functional relationship.*

We have toned down this conclusion to read: “Taken together, these dFISH and in silico data suggest that planarian stem cells adjacent to the CNS are one of the cellular targets of brain-derived *hh* and Wnt signals.”